# A Facile and Efficient Protocol for Preparing Residual-Free Single-Walled Carbon Nanotube Films for Stable Sensing Applications

**DOI:** 10.3390/nano9030471

**Published:** 2019-03-21

**Authors:** Florin Loghin, Almudena Rivadeneyra, Markus Becherer, Paolo Lugli, Marco Bobinger

**Affiliations:** 1Chair of Nanoelectronics, Technical University of Munich, 80333 Munich, Germany; florin.loghin@tum.de (F.L.); markus.becherer@tum.de (M.B.); 2Pervasive Electronics Advanced Research Laboratory (PEARL), Department of Electronics and Computer Technology, University of Granada, 18071 Granada, Spain; 3Faculty of Science and Technology, Free University of Bolzano, 39100 Bolzano-Bozen, Italy; paolo.lugli@unibz.it

**Keywords:** carbon nanotubes, CNTs, carboxymethyl cellulose, CMC, spraying, dispersions, sensing

## Abstract

In this article, we report on an efficient post-treatment protocol for the manufacturing of pristine single-walled carbon nanotube (SWCNT) films. To produce an ink for the deposition, the SWCNTs are dispersed in an aqueous solution with the aid of a carboxymethyl cellulose (CMC) derivative as the dispersing agent. On the basis of this SWCNT-ink, ultra-thin and uniform films are then fabricated by spray-deposition using a commercial and fully automated robot. By means of X-ray photoelectron spectroscopy (XPS), Fourier-transform infrared spectroscopy (FTIR), and scanning electron microscopy (SEM), we show that the CMC matrix covering the CNTs can be fully removed by an immersion treatment in HNO_3_ followed by thermal annealing at a moderate temperature of 100 °C, in the ambient air. We propose that the presented protocols for the ink preparation and the post-deposition treatments can in future serve as a facile and efficient platform for the fabrication of high-quality and residual-free SWCNT films. The purity of SWCNT films is of particular importance for sensing applications, where residual-induced doping and dedoping processes distort the contributions from the sensing specimen. To study the usability of the presented films for practical applications, gas sensors are fabricated and characterized with the CNT-films as the sensing material, screen printed silver-based films for the interdigitated electrode (IDE) structure, and polyimide as a flexible and robust substrate. The sensors show a high and stable response of 11% to an ammonia (NH_3_) test gas, at a concentration of 10 ppm.

## 1. Introduction

Carbon nanotubes (CNTs) have sparked the interest of many researchers in the last two decades because of their outstanding mechanical and electrical properties, which have been comprehensively summarized in previous review papers [1,2,3]. Some of these properties include their small diameter dimensions of only a few nm [4], their mechanical strength [5,6,7], and their high charge carrier mobility [8,9]. Thanks to these intrinsic features, many different applications have been proposed based on CNTs. These applications range from sensors [10,11] to energy storage [12,13] and conductive composites [14,15]. Besides their metallic or semiconducting character, CNTs can be classified in two other categories: single-walled carbon nanotubes (SWCNTs) and multi-walled carbon nanotubes (MWCNTs). The former consists of a single graphite sheet seamlessly wrapped into a cylindrical tube, while the latter comprises an array of such nanotubes that are concentrically nested like the rings of a tree trunk [8]. For sensing applications, SWCNTs promise a higher sensitivity compared with MWCNTs, which can be ascribed to their increased surface-to-volume ratio, and thus an increased number of unsaturated bonds that can act as reaction sites for the sensing specimen [16]. From an economic point of view, the increased sensitivity goes along with an increased product price, as the yield and the purity of SWCNT synthesis protocols is commonly lower than the ones for MWCNTs [17,18]. The advances in the CNT synthesis, purification, and chemical modification enable the integration of CNTs in thin-film electronics and large-area coatings [19]. However, one of the main limitations for the effective and extensive employment of CNTs remains the preparation of stable and high-quality dispersions of CNTs that intrinsically tend to agglomerate and form bundles. These bundles can pose a problem for the deposition of thin-films by clocking the nozzle of the spray gun or inkjet head, affecting the sensing properties, or even drastically changing electrical transport behaviour. In order to avoid this aggregation, many different solvents, reactants, and treatments have been explored [14,20]. The tested surface modifications can be distinguished as follows: (i) mechanical surface modification [21,22,23]; (ii) covalent surface modification [24,25,26]; (iii) non-covalent surface modification [27,28,29,30]; and (iv) irradiation-induced surface modification [31,32]. An extremely promising, non-destructive, and scalable purification procedure based on a carboxy methyl cellulose (CMC) dispersing agent for SWCNTs [33,34,35,36] at a markedly high purity of >95 wt% has recently been reported [19,37]. In these works, the following fabrication procedure was reported: (i) annealing of the raw (as-purchased) SWCNTs in air at a temperature of 350 °C for a duration of 60 min, (ii) dispersion of the SWCNTs in an aqueous solution of CMC followed by sonication, and iii) removal of the CMC and metal catalysts using concentrated hydrochloric acid (HCl). The use of CMC as the dispersing agent not only enables the untangling of the SWCNT bundles, but also provides protection against defects generated by the annealing and acid treatments [37]. However, these studies employ an aggressive halide-containing acid and a high annealing temperature that prohibits the use of most polymer substrates, except for some heat-resistant polyimides.

In this work, we employ CMC as the dispersing agent for SWCNTs and study the effect and the efficiency of three post-deposition treatments on the chemical composition of the SWCNT thin-film. As an addition to previous works, we have found that an immersion treatment of the as-deposited films in nitric acid (HNO_3_) efficiently removes the embedding CMC matrix. The subsequent annealing treatment under ambient conditions at a mild temperature of 100 °C for a duration of 1 h efficiently removes the residuals of the nitric acid treatment. The final film is composed of a well-dispersed and high-quality residual-free SWCNT network that can serve as a starting material for many applications. As one possible application, where the purity of the SWCNT film plays a key role, we propose to test the presented films as bio- or gas sensors.

## 2. Materials and Methods

### 2.1. Preparation of the Spray Ink

In order to disperse CNTs in an aqueous solution, a high molecular weight cellulose derivative, sodium carboxymethyl cellulose (CMC), is used. This kind of dispersant has been previously reported as an excellent agent for dispersing CNTs in water [37]. Further discussion about the effects of different kinds of dispersants on the prepared CNT solution can be found in the work of [38]. CMC is dissolved in deionized (DI) water at a weight content of 0.5 wt%. The solution is then stirred overnight in order to uniformly dissolve the dispersant in water. After this, 18 mg of CNTs is added to 60 g of the 0.5 wt% CMC–aqueous solution to prepare the final 0.03 wt% CMC-based aqueous CNT dispersion. Sonication of the CMC-based solution is performed for 20 min using the probe sonicator Branson 450 Digital Sonifier from Branson Ultrasonics Corporation (Danbury, CT, USA), at a power of 200 W. After the sonication, the solution is centrifuged for a duration of 90 min at a rotation speed of 15 krpm. Finally, 80% of the centrifuged solution is taken from top and used for the deposition [39].

### 2.2. Spray Deposition and Post Treatments

The CNT films presented here were deposited with the aid of an air atomizing nozzle. Our automated setup consists of an industrial low volume low pressure (LVLP) air atomizing spray valve 781S from Nordson Corporation (Westlake, OH, USA) with a full cone profile in combination with an automated motion platform from Werner Wirth GmbH (Hamburg, Germany) and a proportional-integral-derivative (PID) controlled aluminum hot plate [40]. The selection of the spray parameters as well as the protocol for the preparation of the inks are the result of a long lasting tailoring and were subject to previous publications [41,42,43,44]. The nozzle is operated in a wet spraying regime, while the sample is heated. A chemical post-deposition treatment is required to remove the CMC-matrix that embeds the CNTs. The samples were placed in 1:4 HNO_3_/Di-H_2_O for a duration of 12 h. This step converts the nature of the network from insulating to conducting. After the immersion treatment, some SWNCT films were annealed at a temperature of 100 °C for a duration 1 h.

### 2.3. Scanning Electron Microscopy

Field-emission scanning electron microscope (FESEM) images were recorded using an NVision40 from Carl Zeiss (Oberkochen, Baden-Wurttemberg, Germany) at an acceleration voltage of 7.0 kV and an extraction voltage of 5.0 kV. The working distance was adjusted in a range of 5–6 mm to achieve the best image quality.

### 2.4. X-Ray Photoelectron Spectroscopy

X-ray photoelectron spectroscopy (XPS) measurements were performed at a base pressure of 5 × 10^−10^ mbar with a monochromatic aluminium $K_{\alpha }$ anode as x-ray source, at an operating power of 350 W. The high-resolution spectra were acquired using a SPECS Phoibos hemispherical analyser from SPECS Surface Nano Analysis GmbH (Berlin, Germany) at a pass-energy of 20 eV with an energy resolution of 0.05 eV. As described in previous publications [45,46], the raw data were processed using the software CasaXPS from Casa Software Ltd. (Teignmouth, UK).

### 2.5. Sheet Resistance Measurement

The sheet resistances were measured using a four-point probe head from Jandel (Linslade, UK) connected to a B2901A Keysight (Santa Rosa, CA, USA) source measuring unit (SMU). A constant current of 10 µA was sourced for all measurements.

### 2.6. Transmission Measurement

The transmittance spectra were recorded in the visible range using a 300 W xenon arc lamp, chopped at a frequency of 210 Hz. The light passes through an Oriel Cornerstone 260¼ m monochromator from Newport Corporation (Irvine, CA, USA) and a silicon-based photodiode with a transconductance amplifier connected to a 70105 Oriel Merlin digital lock-in amplifier from Newport Corporation. The calibration of the photodiode was performed with a glass substrate to determine the pure transmission of the CNT films.

### 2.7. Fourier-Transform Infrared Spectroscopy

Fourier-transform infrared spectroscopy (FTIR) measurements were recorded using an ALPHA II spectrometer from Bruker (Billerica, MA, USA) equipped with a platinum attenuated total reflection (ATR) module. Polished silicon was used as the substrate. The spectra were recorded in a wavenumber region of 400–4000 cm^−1^, at a resolution of 2 cm^−1^.

### 2.8. Gas Measurement Setup

For the characterization of the CMC-CNT films as gas sensors, the sample was mounted onto a module that consists of a Peltier element used for the temperature control, a Pt100 thermoresistor for in situ temperature monitoring, and leads for the contacting of the sensor (see Appendix A for a photograph of the sensor module). The holder was inserted into a home-made gas chamber, into which a nitrogen carrier gas and different test gases can be fluxed. The sensor response to the test gas ammonia (NH_3_) was characterized by exposing the sample to various concentrations. To maintain a constant flux, the carrier gas (N_2_) was mixed with NH_3_ to generate different concentrations, as shown in the flux profile over time in Appendix A (see Appendix A).

### 2.9. Screen Printing of the Sensor Electrodes

The CMC-CNT films were contacted electrically using the silver-based screen print paste Loctite1010 from Loctite (Hartford, CT, USA). The films were screen printed using an inexpensive manual screen-printing machine FLAT-DX 200 from Siebdruck Versand (Magdeburg, Germany) and a 165T (165 threads per cm) polyester-based mesh. After printing, the electrodes were dried in a UF55 oven from Memmert (Schwabach, Germany) at a temperature of 60 °C for a duration of 15 min.

## 3. Results and Discussion

### 3.1. Film Characterization

The effect of the immersion treatment in HNO_3_ solution was studied by means of XPS to resolve the chemical alterations and by means of SEM to observe the change in morphology. In detail, the following treatments, labelled I, II, and III, were investigated: I. as deposited; II. immersion treatment in an aqueous 25 wt% HNO_3_ solution for a duration of 12 h; and III. treatment II followed by thermal annealing at a temperature of 100 °C on a hot plate for a duration of 1 h, in the ambient air. The XPS survey scans of the CNT films deposited onto polished silicon (Si) substrate with a natural oxide layer before and after immersion in HNO_3_ solution, as well as after the thermal annealing treatment, are depicted in Figure 1a alongside the SEM images for the treatments (b) I, (c) II, and (d) III. The SEM images indicate that a densification of the CNT-film is induced by the immersion in HNO_3_, while there is no visible change from treatment II to III.

From the XPS survey scans shown in Figure 1a, it can be recognized that the sodium Na 1s core-level peak and the lower binding energy O 1s-related peak vanish after the immersion treatment, which is accompanied by the appearance of a low Si-induced signal for treatment II and III. This observation clearly indicates that the CMC matrix covering the CNT film is removed, and thus a small portion of the bare Si substrate is revealed. The structural formula for a CMC monomer that contains sodium atoms with the chemical formula $C_{28}H_{30}{\mathrm{Na}}_{8}O_{27}$ is depicted in Appendix A (see Appendix A). Further, because of the immersion in HNO3, a nitrogen induced peak with a low signal occurs after treatment II. In agreement with the well-resolved high-resolution scans depicted in Figure 2, the nitrogen peak vanishes after the annealing treatment III, which proves that the nitric acid residuals can be removed by evaporation. The high resolution spectra for the relevant core-level transitions were normalized with regard to their relative sensitivity factors (RSF) (see Appendix A for a summary) [47,48] and are depicted in Figure 2 for (a) C 1s, (b) O 1s, (c) Na 1s, and (d) N 1s. From these spectra, following conclusions can be drawn: (i) the shapes of the C 1s and O 1s peaks change, which indicates that the prominent carbon- and oxygen-containing compounds are altered after the immersion treatment; (ii) the sodium peak vanishes completely, which is a proof for the efficient removal of CMC; and (iii) the nitrogen contribution decreases after the thermal annealing treatment, which indicates that the trace amount of HNO_3_ on the CNT film after the immersion treatment can be removed by evaporation. A summary of the elemental concentrations for carbon, oxygen, sodium, silicon, and nitrogen that are determined from the XPS core-level spectra and the RSFs is given in Table 1.

Next, the chemical shapes of the immersion treatment will be studied in more detail by looking to the high-resolution C 1s spectra that are illustrated in Figure 3 for treatment (a) I, (b) II, and (c) III. The raw spectra were deconvoluted into six different contributions, in accordance with the chemical formula of CMC and the literature [49,50,51]. The contributions are labelled from C1–C6 and can be associated with different hybridizations of the carbon atoms in the CMC matrix and in the CNT film, as follows: C1s is mainly composed of sp^2^ hybridized carbon atoms that are characteristic for CNTs and graphene, C2 represents sp^3^ hybridized carbon that is present in graphite or organic molecules such as CMC, C3–5 denote carbon–oxygen compounds that are present in CMC or oxidized CNTs, and C6 denotes the $\pi -\pi^{*}$ transition. The relative contributions that sum up to the overall C 1s signal are summarized in the table shown in Figure 3d. It can be seen that the immersion treatment leads to the removal of O–C–O and O–C=O groups that are present in the CMC molecule, whereas the relative contribution for sp^2^ hybridized carbon increases significantly from around 0% for the as-deposited film to around 34% for the film after immersion treatment. This result further evidence that, for treatment I, the CMC matrix entirely covers the CNT network and shields it from the XPS measurement, whereas after treatment II, CMC is removed and the pure CNT-film is revealed. As the CNT film in its pristine form is known to be largely composed of sp^2^ hybridized carbon atoms, the sp^2^-percentage drastically increases after treatment II. The remaining sp^3^ hybridized carbon atoms that should not be present in an ideal CNT can be ascribed to carbon-containing contamination and carbon–oxygen compounds in the nanotube [52,53].

Only a small difference can be recognized for the shape of the C 1s signal for treatment II and III. The removal of carbon–oxygen compounds that are mostly associated with C–O groups can eventually be attributed to the removal of moisture on the CNT film by evaporation. It should be noted that, similar to other groups, we have focused the main discussion of the high resolution scans on the C 1s spectra because the O 1s signal is usually superimposed by contamination and a thin water film that both add wide-band contributions to the spectra and partially obscure the alterations of the treatments [54]. Nevertheless, similar conclusions as for the C 1s spectra shown in Figure 3a–c can be drawn from the O 1s spectra illustrated in Figure 2b. Before the immersion treatment, the O 1s spectra is composed of a peak centred around a binding energy of 533 eV and higher binding energy contributions. The peak at 533 eV can be ascribed to a thin water film [55,56] that is present on all samples, whereas the higher binding energy contributions are associated with the oxygen atoms in the CMC matrix (see Appendix A for the chemical structure of the CMC molecule). Besides XPS, the effect of the treatments has been studied using transmission as well as Fourier-transform infrared spectroscopy under attenuated total reflection (ATR). The as-deposited films that are later also used for gas sensors show a remarkably high optical transmission of around 96% at a wavelength of 550 nm (see Appendix A). However, because of the high transparency and the accuracy of the measurement, no difference can be seen for the different treatments. From the FTIR absorption spectra shown in Figure 4, a clear change can be identified between treatment I and the other two treatments including the silicon reference spectrum. Before the HNO_3_ immersion, the spectrum is dominated by the contribution from CMC that gives rise to absorption peaks centred around wavenumbers of 1718, 1223, 1220, 1090, and 1024 cm^−1^, which are labelled consecutively from 1–4 in the graph. In agreement with the literature, these peaks are associated with the asymmetrical and symmetrical COO stretching modes, as well as the C–H bending mode, respectively [57,58]. After immersion, that is, for treatment I and II, the spectra cannot be distinguished from the silicon reference spectrum with its characteristic S–H-induced peaks around 2327 and 2113 cm^−1^, denoted as 5 and 6 in the graph, anymore. As the XPS and the SEM measurements have already shown that the CNT film is not removed by the immersion treatment, this result further proves that the embedding CMC matrix is efficiently removed after treatment II. After treatment II, as well as after treatment III, no difference in the spectrum compared with the silicon reference can be seen as the surface sensitivity of this optical technique is known to be too low for thin layers.

In summary, the XPS and the FTIR measurements show that the immersion in HNO_3_ solution efficiently removes CMC, whereas the subsequent thermal annealing step removes residual HNO_3_ on the film. The presented protocol essentially allows for the use of such deposited films in a wide range of applications because of the residue-free removal of the dispersant, which results in a pristine SWCNT film. We propose that this protocol can serve as a platform for future CNT-based sensors that to date are already used in, for example, gas, temperature, humidity, and biosensors [59]. To prove the excellent usability of the presented CNT films for sensing application, in the next section, the films are characterized as gas sensors to detect ammonia.

### 3.2. Sensing Application

To demonstrate the effectiveness of the discussed post-deposition treatments, the CNT films have been used as gas sensors that show a stable and high response to NH_3_. For the CNT sensing films, the same parameters described in Section 2 were applied and polyimide (Kapton^®^HN from DuPont, Wilmington, DE, USA) was used as a flexible, as well as mechanically and thermally robust, substrate. After the deposition and the post-treatments, a sheet resistance in a range of 1–10 MΩ/sq. was measured. Similar to a previous work [60], the CNT films were contacted using a highly conductive silver-flake-based [61] and screen-printed interdigitated electrode (IDE) structure, as illustrated in the photograph in the inset in Figure 5b (see Section 2.8 for the experimental details). The thickness and the sheet resistance of the silver electrodes were 4.4 ± 0.1 µm and 139 ± 8 mΩ/sq., respectively, which yields to a resistivity of 6.1 × 10^−7^ Ω∙m. This resistivity is increased by a factor of around 38 compared with the value for bulk silver. The discrepancy in the resistivity can be ascribed to residuals of the solvent and high flake-to-flake resistance [62,63]. As described in Section 2.8, the CNT-based gas sensors were subjected to ammonia diluted with nitrogen at different concentrations of 0, 10, 20, 30, 50, and 80 ppm, respectively. These concentrations lie well in the sensitivity range of CNT-based sensors that is reported from a lower limit of detection of around 40 parts per billion (ppb) up to around a value of 1000 ppm, for NO_2_ gas [64]. The change in resistance of the sensor over time is depicted in Figure 5a for increasing concentrations that are labelled consecutively from 0 to 5 in the graph. The slow increase in resistance for each concentration, which occurs after a saturation behavior, arises as a result of an active recovery step that is required for the desorption of the sensing specimen. From these transients, it can be seen that the resistance of the CNT film is sensitive to the ammonia test gas because of charge-transfer processes at the CNT–NH_3_ interface that dope or dedope the CNT-film and, therefore, lead to a change in the charge carrier density and resistance. This effect is known and has been widely discussed in the literature [10,65]. It should be noted that, so far, these results just show the sensitivity of the CNT-films for NH_3_ gas. For many applications, it is important to identify an unknown test gas, which requires the sensor to be chemically selective. A good degree of selectivity was presented by Star et al. for metal-decorated CNT-based sensors [66]. The normalized response of the gas sensor with regard to the resistance value at 0 ppm test gas, that is, *R*_0_, is depicted in Figure 5b for two measurement cycles. The sensor shows an increase in response with NH_3_ concentration that exhibits an almost linear behavior until a concentration of around 30 ppm and saturation above. At 30 ppm, the sensor shows a high response of around 18% that reduced down to 15% for the second cycle. This reduction in response can be ascribed to an incomplete removal of adsorbed ammonia specimen during the active recovery cycle. In agreement with the findings of other authors working in the same field, the slow and incomplete recovery of CNT-based gas sensors is a drawback that remains until today. Besides using active recovery, numerous alternative methods were applied to recover the sensitivity of the CNT films. These methods include (i) fluxing with nitrogen [67] or argon [68] for a duration up to 10 h; (ii) thermal annealing in ambient air at a temperature of 200 °C, for a duration of 1 h [68]; and (iii) the use of ultraviolet (UV) light [10,69] for around 10 min and the evacuating to high vacuum at a temperature of 500 k for several hours [70]. Attributed to the low power consumption, ease-of-use, and short duration, in this work, we have utilized active recovery as the method of choice. Nevertheless, it should also be noted that this technique does not completely remove the adsorbed specimen and, for inhomogeneous films, can lead to the formation of hot spots or hot areas.

## 4. Conclusions

We report on the preparation of SWCNT thin-films by spray deposition, using CMC as the dispersant. Further, we analyze the effect of the different post-treatments with regard to the removal of the dispersants. In particular, the immersion in HNO_3_ efficiently removes the CMC-matrix. Additionally, the thermal annealing at 100 °C facilitates the desorption of residual HNO_3_ on the film, allowing for a residue-free removal, which results in a pristine CNT film. This extra step enhances the purity of the spray deposited CNT layers and enables the employment of such films in a variety of applications including sensing as a result of a high level of purity.

## Figures and Tables

**Figure 1 nanomaterials-09-00471-f001:**
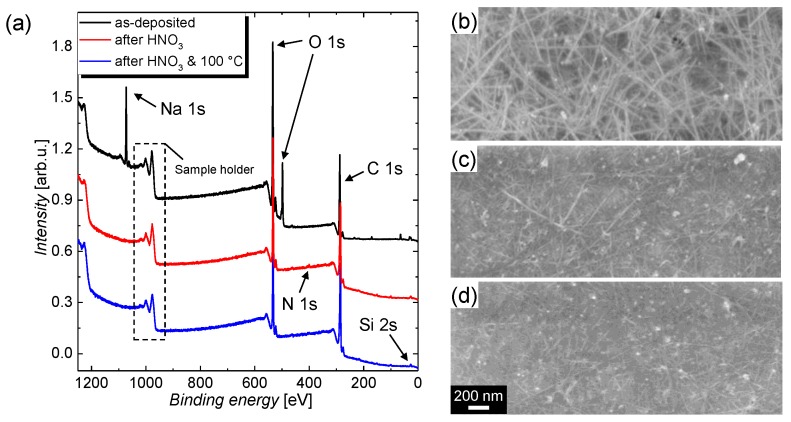
(**a**) X-ray photoelectron spectroscopy (XPS) survey scans of (top black curve) an as-deposited carboxymethyl cellulose (CMC)-carbon nanotube (CNT) film (treatment I), (middle red curve) a CMC-CNT film after immersion in HNO_3_ (treatment II), and (bottom blue curve) a CMC-CNT film after treatment II followed by thermal annealing at a temperature of 100 °C (treatment III). The peaks that are associated with the different elements are indicated in the spectra. Scanning electron microscope (SEM) images recorded for CMC-CNT films subjected to the treatments (**b**) I, (**c**) II. and (**d**) III. The scale bar in (**d**) applies to all SEM images.

**Figure 2 nanomaterials-09-00471-f002:**
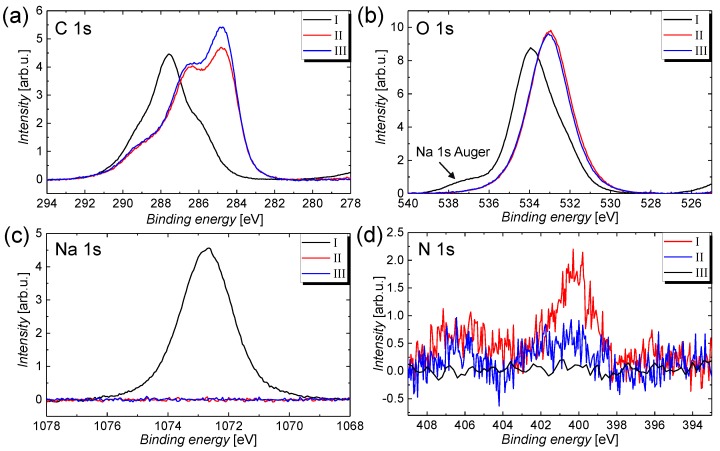
High-resolution spectra for the different treatments and the core-level spectra (**a**) C 1s, (**b**) O 1s, (**c**) Na 1s, as well as (**d**) N 1s. The spectra are normalized with respect to the concentrations of the specific elements.

**Figure 3 nanomaterials-09-00471-f003:**
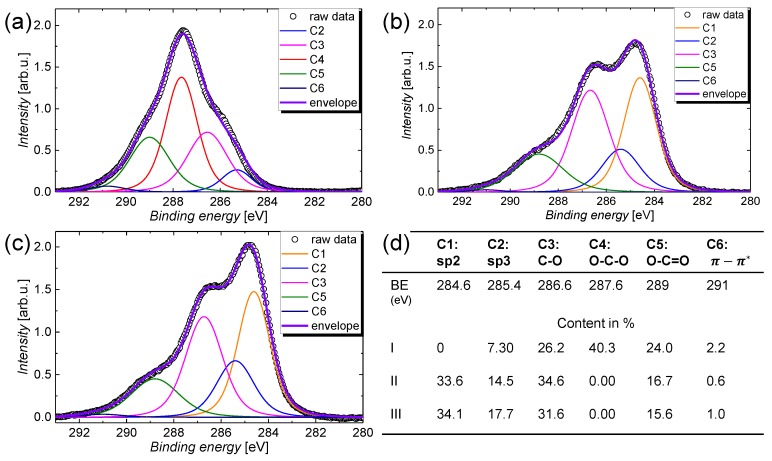
High resolution C 1s XPS spectra for the CMC-CNT films that were subjected to different treatments: (**a**) I, (**b**) II, and (**c**) III. The raw spectra were decomposed into different contributions by Lorentzian fits. (**d**) Table that summarizes the different contributions of carbon-containing species that add up to the spectra (**a**–**c**). BE—binding energy.

**Figure 4 nanomaterials-09-00471-f004:**
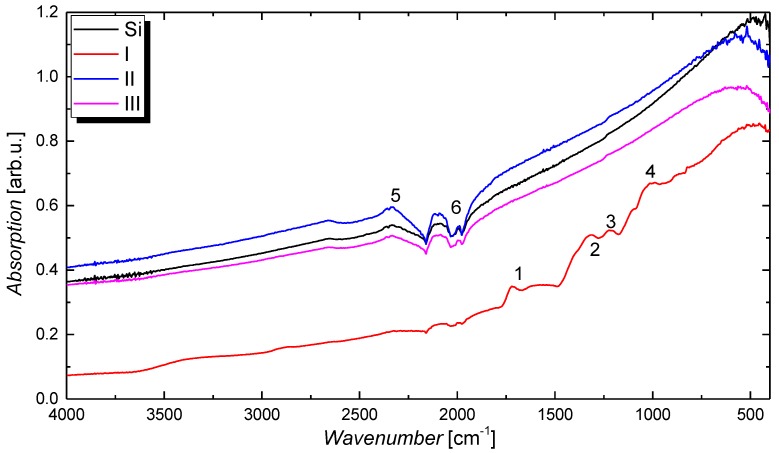
Fourier-transform infrared spectroscopy (FTIR) absorption spectrum for CMC-CNT films on silicon substrate subjected to the different treatments I–III. The plot includes the reference spectrum of the silicon substrate, as well as the consecutive numbering of absorption peaks that is discussed in the main text.

**Figure 5 nanomaterials-09-00471-f005:**
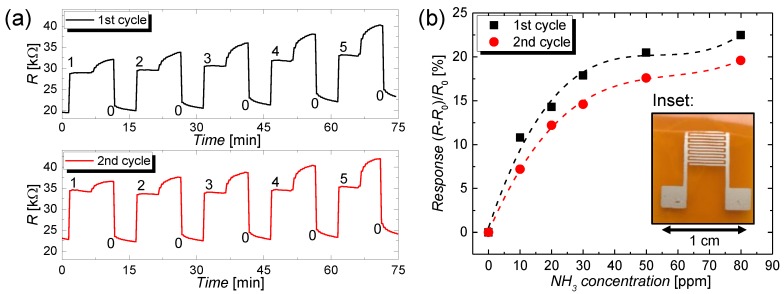
(**a**) Increase in resistance over time of a CNT-based sensor with a screen printed interdigitated electrode (IDE) structure that is shown in the inset in (**b**). The sensor is subjected to NH_3_ gas for two cycles and for increasing concentrations of 0, 10, 20, 30, 50, and 80 ppm. The moments at which the different concentrations are input are labelled consecutively in the graph from 0–5. (**b**) Saturated responses as a function of the different concentrations for two sensing cycles.

**Table 1 nanomaterials-09-00471-t001:** Relative elemental composition of the carboxymethyl cellulose (CMC)-carbon nanotube (CNT) films for different post-treatments. The compositions were determined via X-ray photoelectron spectroscopy (XPS) and considering the relative sensitivity factors for the different peaks. The details for the treatments can be found in the main text.

Treatment	C 1s	O 1s	Na 1s	Si 2s	N 1s
Content in %					
I	50.6	45.4	3.70	0.30	0
II	63.2	34.5	0.0	0.55	1.75
III	65.3	33.4	0.0	0.73	0.58

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
