# Peer review of "A Facile and Efficient Protocol for Preparing Residual-Free Single-Walled Carbon Nanotube Films for Stable Sensing Applications"

_nanomaterials, 2019, doi:10.3390/nano9030471_

Round 1
Reviewer 1 Report
The authors have addressed the questions raised by the reviewers.
Author Response
We thank the reviewer for providing a fast feedback for our submission. The reviewer did not raise any further questions in this revision round.
Reviewer 2 Report
The manuscript describes the preparation of single-walled carbon nanotube (SWCNT) films with the aid of carboxymethylcellulose (CMC) as dispersing agent. The dispersions are used for the preparation of gas sensors for the detection of ammonia (NH3). Authors have revised the manuscript taking into account the comments of the reviewers, addressing the main concerns and improving the quality of the manuscript, which could be published after minor revision:
1) It is possible to use the sensor for quantitative analysis of NH3? The analytical features of the method should be calculated and shown, e.g. linear range, limits of detection and quantification, precision of the measurements in terms of relative standard deviation (RSD, %), etc.
2) It is mentioned that in the second cycle the response is reduced due to an incomplete removal of NH3. Thus, is it not possible to reuse the sensors? Is there any post-treatment that could remove such NH3 so that the sensor can be further used without losing sensitivity?
Author Response
Response to the Reviewer
We thank the Reviewer for his/her valuable feedback for our submission to MDPI Nanomaterials. We have done any effort to address the comments of the Reviewer. Please find our responses to the reviewer below. The comments of the reviewer are written in bold text.
The manuscript describes the preparation of single-walled carbon nanotube (SWCNT) films with the aid of carboxymethylcellulose (CMC) as dispersing agent. The dispersions are used for the preparation of gas sensors for the detection of ammonia (NH3). Authors have revised the manuscript taking into account the comments of the reviewers, addressing the main concerns and improving the quality of the manuscript, which could be published after minor revision:
1) It is possible to use the sensor for quantitative analysis of NH3? The analytical features of the method should be calculated and shown, e.g. linear range, limits of detection and quantification, precision of the measurements in terms of relative standard deviation (RSD, %), etc.
Response to the reviewer: In the revised manuscript, marked in green, we have included a brief discussion of the detection limits of CNT-based sensors as well as for the chemical selectivity. In detail, we write:
These concentrations lie well in the sensitivity range of CNT-based sensors that is reported from a lower limit of detection of around 40 parts per billion (ppb) up to around a value of 1000 ppm, for NO2 gas [1]. It should be noted that, so far, these results just show the sensitivity of the CNT-films for NH3 gas. For many applications, it is important to identify an unknown test gas, which requires the sensor to be chemically selective. A good degree of selectivity was presented by Star et al. for metal-decorated CNT-based sensors [2].
References:
1. Zaporotskova, I. V.; Boroznina, N.P.; Parkhomenko, Y.N.; Kozhitov, L. V. Carbon nanotubes: Sensor properties. A review. Mod. Electron. Mater. 2017, 2, 95–105.
2. Star, A.; Joshi, V.; Skarupo, S.; Thomas, D.; Gabriel, J.C.P. Gas sensor array based on metal-decorated carbon nanotubes. J. Phys. Chem. B 2006, 110, 21014–21020.
2) It is mentioned that in the second cycle the response is reduced due to an incomplete removal of NH3. Thus, is it not possible to reuse the sensors? Is there any post-treatment that could remove such NH3 so that the sensor can be further used without losing sensitivity?
Response to the reviewer: In the revised manuscript, marked in green, we have included a discussion and literature review for the recovery of CNT-based sensors. We have also noted the disadvantage of the active recovery method. In detail, we write: Besides using active recovery, numerous alternative methods were applied to recover the sensitivity of the CNT films. These methods include i) fluxing with nitrogen (1) or argon (2) for a duration up to 10 h, ii) thermal annealing in ambient air at a temperature of 200 °C, for a duration of 1 h (2), iii) the use of ultraviolet (UV) light (3,4) for around 10 min and the evacuating to high vacuum at a temperature of 500 k for several hours 5. Attributed to the low power consumption, ease-of-use and short duration, in this work, we have utilized active recovery as the method of choice. Nevertheless, it should also be noted that this technique does not completely remove the adsorbed specimen and, for inhomogeneous films, can lead to the formation of hot spots or hot areas.
and provide following references:
1. Nguyen, H.Q.; Huh, J.S. Behavior of single-walled carbon nanotube-based gas sensors at various temperatures of treatment and operation. Sensors Actuators, B Chem. 2006, 117, 426–430.
2. Kong, J.; Franklin, N.R.; Zhou, C.; Chapline, M.G.; Peng, S.; Cho, K.; Dai, H. Nanotube molecular wires as chemical sensors. Science (80-. ). 2000.
3. Li, J.; Lu, Y.; Ye, Q.; Cinke, M.; Han, J.; Meyyappan, M. Carbon nanotube sensors for gas and organic vapor detection. Nano Lett. 2003.
4. Lu, Y.; Li, J.; Han, J.; Ng, H.T.; Binder, C.; Partridge, C.; Meyyappan, M. Room temperature methane detection using palladium loaded single-walled carbon nanotube sensors. Chem. Phys. Lett. 2004.
5. Tsai, M.H.; Lin, H.M.; Tsai, W.L.; Hwu, Y. Examine the gas absorption properties of single wall carbon nanotube bundles by X-ray absorption techniques. Rev. Adv. Mater. Sci. 2003.

This manuscript is a resubmission of an earlier submission. The following is a list of the peer review reports and author responses from that submission.
Round 1
Reviewer 1 Report
The manuscript describes the preparation of single-walled carbon nanotube (SWCNT) films with the aid of carboxymethylcellulose (CMC) as dispersing agent. Spray-deposition is used for the fabrication of the films. The topic is potentially interesting; however, the main results and discussion has been focused on the effect of HNO3 immersion and subsequent annealing, while little information regarding e.g. mechanical and physical properties of the films itself is provided. Authors should provide more information and further discuss some of the points before this manuscript could be publishable.
The following suggestions are given for improving the manuscript:
1) The conditions for the preparation of the spray ink, were they taken from previous publications? (in that case please indicate appropriate reference). Otherwise, were the amount of CNT, wt% of CMC, etc. optimized to obtain the best dispersion? Or how were such conditions selected?
2) As previously advanced, the results and discussion section is directly focused in evaluation of the removal of CMC after two treatment. However, authors should first prove the effect of CMC on the dispersion of CNTs at such conditions and also characterized the deposited films in terms of mechanical, physical properties, etc. before and after the treatments. More information regarding the films are needed.
3) Figure 1(a) should also depict the XPS spectrum after step II, that is immersion in HNO3but without the thermal annealing. The same for the SEM images. Later on XPS differences from treatment I, II and III are discussed for C1s, the same should apply to XPS survey scan and SEM images.
4) Figure 1(a): all the peaks shown in the XPS survey scans should be assigned and labeled within the spectrum.
5) Only C1s region is discussed, authors should also discuss the effects observed in the O1s regions upon the different treatments.
6) Table in fig. 2(d) shows significant changes in the sp2and sp3 content of the samples after the different treatments. These changes (e.g. the fact that at stage I sp2hybridization was 0% and after III is 34.1%) should be explain and discussed in more detail. It is mentioned, but the reason behind this observation should be further discussed.
7) The sentence in lines 196-197 “Authorship must be limited….” should be removed.
8) Funding and acknowledgements are almost similar.
9) There are related references about CNTs and CMC that are not mentioned, e.g.:
- Riou et al. J Nanosci Nanotechnol. 2009 Oct;9(10):6176-80.
- Hajian et al. Composites Science and Technology, Volume 159, 3 May 2018, Pages 1-10
- Young et al. Single-walled carbon nanotube (SWNT)-carboxymethylcellulose (CMC) dispersions in aqueous solution and electronic transport properties when dried as thin film conductors, (2018) Journal of Dispersion Science and Technology, DOI: 10.1080/01932691.2018.1452759
- Liu et al. RSC Adv., 2016,6, 67260-67270
10) The manuscript is not too long and there are only two figures and one table, thus, instead of placing figures S1 and S2 as well as table S1 in a supplementary file, they could be placed within the main text.
Reviewer 2 Report
This manuscript reports a method to prepare a residual-free SWCNT films, by using CMC as a dispersing agent, HNO3 as a washing agent, and heat treatment for HNO3/H2O removal. XPS and SEM were used for analyzing the samples. It surely does appear that elemental compositions are affected by the treatment. However, the results do not seem to be the subject of the urgent publication as a communication and the manuscript is rather premature for publication at this stage. There seems to be more than SWCNT after the sequential treatment. Other analytic methods, such as UV/Vis, IR, Raman, TGA, and so on, should be performed to verify the results. Because CNT is electrically active, comparison of dc conductivity is also required. In addition, although the authors mentioned sensor applications in a few times in the manuscript, there is no single experiment related to the applications.